# The Postharvest Application of Carvone, Abscisic Acid, Gibberellin, and Variable Temperature for Regulating the Dormancy Release and Sprouting Commencement of Mini-Tuber Potato Seeds Produced under Aeroponics

**DOI:** 10.3390/plants12233952

**Published:** 2023-11-24

**Authors:** Tiandi Zhu, Huaidi Pei, Zhongwang Li, Minmin Zhang, Chen Chen, Shouqiang Li

**Affiliations:** 1Biotechnology Institute, Gansu Academy of Agricultural Sciences, Lanzhou 730070, China; ztd@gsagr.ac.cn (T.Z.); phd20100202@gsagr.cn (H.P.); zwli-biotech@gsagr.cn (Z.L.); zhangmm@gsagr.cn (M.Z.); sjschench@gsagr.ac.cn (C.C.); 2Agricultural Product Storage and Processing Research Institute, Gansu Academy of Agricultural Sciences, Lanzhou 730070, China; 3Gansu Innovation Center of Fruit and Vegetable Storage and Processing, Lanzhou 730070, China

**Keywords:** mini-tuber potato, dormancy, sprouting, carvone, abscisic acid, gibberellin, variable temperature

## Abstract

This study investigated the efficacy of carvone, abscisic acid (ABA), gibberellin (GA3), and variable temperature in managing dormancy and sprouting in aeroponically grown mini-tuber potato (*Solanum tuberosum* L.) seeds. The results showed that carvone treatment effectively reduced the weight loss rate by 12.25% and decay rate by 3.33% at day 25 compared to control. ABA treatment significantly enhanced the germination rate, increasing it to 97.33%. GA3 treatment resulted in the longest sprouts of 14.24 mm and reduced the MDA content by 23.08% at day 30, indicating its potential in shortening dormancy and maintaining membrane integrity. The variable-temperature treatment demonstrated a balanced performance in reducing weight loss and maintaining a lower relative conductivity, indicating less cellular damage. The enzymatic activities of α-amylase, CAT, and SOD were modulated by the treatments, ensuring a balanced enzymatic environment for seed vitality. These results establish a solid basis for improving postharvest management strategies to optimize germination uniformity and preserve the quality of aeroponic potato seeds during extended dormancy, promising enhanced yield and productivity in potato cultivation.

## 1. Introduction

The potato (*Solanum tuberosum* L.), belonging to the Solanaceae family, is one of the four most important food crops globally. Its significance in global food production is underscored by its high nutritional content and prolific yield [1,2,3]. Due to the increasing focus on utilizing germplasm resources, it is critical to develop efficient storage methods for seed potatoes. In potato cultivation and production, seed potatoes play a crucial role due to their direct influence on both yield and quality, serving as the linchpin of potato production [4,5]. During storage, seed potatoes undergo intricate physiological and biochemical changes, resulting in changes to their chemical composition. The conditions of storage have a direct bearing on their vitality and quality [6]. In particular, initial seed potatoes, characterized by their small size, light weight, and diminished starch and nutrients, risk significant evaporation losses if not carefully managed postharvest [7]. This predisposition could result in reduced germination, stunted seedling growth, and energy reserve depletion, thereby obstructing potato production. Mini-tuber potato seeds cultivated under aeroponics face heightened challenges.

Potato aeroponics, an innovative production method, involves suspending potato roots in hermetically sealed containers. These chambers are intermittently misted with a nutrient-rich solution using specialized devices, ensuring the potatoes receive essential water and nutrients [8,9]. Compared to traditional seed potato farming, aeroponics offers multiple benefits: superior yields of high-quality initial seed potatoes, shorter production cycles, an absence of seasonal constraints, precision in production control, and diminished dependence on tuber seedlings. Yet, potatoes produced through aeroponics tend to exhibit increased tuber moisture, enlarged cuticular pores, and elevated water content, and are more susceptible to pests and diseases [10].

Strategies for managing potato tuber dormancy and sprouting include the use of chemicals and temperature variations [11]. Abscisic acid (ABA) plays a crucial role in dormancy, controlling the status and germination timing to suit varying environmental conditions and survival needs. ABA is fundamental in curtailing growth and development, especially in potatoes, by inhibiting sprouting, prolonging dormancy, and thwarting untimely growth. Additionally, ABA enhances the potato’s resilience against unfavorable conditions [12]. In contrast, gibberellic acid (GA3) has demonstrated effectiveness in shortening dormancy durations. It plays an essential role in seed dormancy, promoting germination, ending dormancy, and interacting with other hormones, significantly impacting seed growth and development [13]. Carvone has a dual function: regulating sprouting and protecting mini-tubers from decay [14]. Among its actions, a singular treatment with minimal carvone doses proves most effective for mini-tuber sprout control. Field tests reveal that carvone treatments do not negatively affect mini-tubers’ emergence, field traits, yield, or harvested potato quality. Carvone can also serve both as a sprout suppressant for commercial potatoes and a sprout inhibitor for seed potatoes [15,16]. Research studies highlight temperature’s significant impact on potato dormancy. Lower temperatures extend dormancy, while higher ones might induce early germination [17]. Tubers generally sprout faster when stored between 8 °C and 12 °C for 5 to 9 months [18]. Given potatoes’ temperature sensitivity, understanding postharvest processes is essential for determining ideal storage conditions. For fresh market potatoes, a temperature range of 3 °C to 5 °C is vital, while processed potatoes should be kept at a minimum of 4 °C to retain flavor and quality [19]. Conversely, elevated temperatures, especially between 16 °C and 30 °C, hasten germination, making sprouted potatoes apt for planting [20]. Therefore, strategic temperature regulation is vital for ideal potato storage and sustainable farming practices.

Dormancy intensity follows recognizable patterns, with original species showing the strongest dormancy, followed by first-generation and subsequent generations. Original species typically exhibit prolonged and variable dormancy phases. The aeroponic storage technique for these potatoes remains in its early stages. This research focuses on the aeroponic growth of these potatoes, investigating postharvest physiochemical alterations. The primary goal is to control dormancy release and initiate sprouting before planting, providing a foundation for aeroponic seed potato cultivation. This investigation aims to enhance seed potato farming methods and bolster sustainable agriculture.

## 2. Materials and Methods

### 2.1. Plant Materials and Treatment

The potato cultivar selected for this study was ‘Atlantic Ocean’, obtained from Dingxi, China, in August 2021. After harvest, these potatoes were immediately transported to the laboratory. Mini-tuber potatoes that displayed uniform growth conditions, consistent size, and showed no evidence of damage or infestations were carefully selected.

These potatoes were subjected to five treatments: (1) Storage at variable temperature (alternating between 4 °C and 20 °C every 7 days, followed by room-temperature storage with scattered light). (2) Treatment with 0.02 mL/kg of carvone, sourced from Wuhan Dahua Weiye Pharmaceutical Chemical Co., Ltd.(Wuhan, China) (3) Treatment with 4 mg/L of abscisic acid (ABA), obtained from Shanghai Yuanye Bio-Technology Co., Ltd. (Shanghai, China) (4) Treatment with a mixture of 10 mg/L of gibberellic acid (GA3) and 1% thiourea solution, acquired from Shanghai Yuanye Bio-Technology Co., Ltd. The potatoes were immersed in each solution for 30 min, followed by treatment with a sealed aerosol for 5 days and subsequent initiation of ventilation. (5) For the control group (CK), potatoes were soaked in water and stored at ambient temperature (20 °C). Each treatment was replicated three times. Various parameters were evaluated at intervals during the study (0, 5, 10, 15, 20, 25, and 30 days): weight loss rate, tuber decay incidence, sprout length, germination rate, relative conductivity, malondialdehyde (MDA) content, α-amylase, SOD, and CAT activities.

### 2.2. Weight Loss Rate

The weight loss rate was determined following a modified methodology outlined by Emragi et al. [21]. At the experiment’s onset, six potatoes were randomly selected from each batch. The initial masses of these potatoes were recorded, followed by subsequent recordings at five-day intervals. Weight loss rate was calculated as:Weight Loss Rate (%) = (Initial Weight − Final Weight)/Initial Weight × 100.

### 2.3. Tuber Decay Incidences

To determine decay incidences, tuber surfaces were visually examined for soft rot symptoms, as described by Nyankanga et al. [22]. At every scheduled interval, a set of 50 tubers were inspected for decay. Decay incidences was calculated as:Tuber Decay Incidences (%) = Number of Decayed Tubers/Total Number of 
Tubers × 100.

### 2.4. Sprout Length and Germination Rate

Germination length and rate were determined following Mahto et al.’s method [23]. Sprout lengths were precisely measured using a measuring tape. Germination rate was calculated as:Germination Rate (%) = Number of Germinated Tubers/Total Number of 
Tubers × 100.

### 2.5. Relative Conductivity and MDA Content

Relative conductivity was measured with minor modifications to the method described by Xu et al. [24]. Ten tissue samples were collected from each group, carefully extracted, rinsed, and then submerged in 20 mL of distilled water for a 30 min duration at a consistently maintained temperature of 25 °C. The initial conductivity (*C*_0_) was measured. Following a cycle of heating and cooling, the final conductivity (*C*_1_) of the samples was recorded. The relative conductivity was subsequently calculated using the formula:Relative Conductivity (%) = (*C*_1_ − *C*_0_)/*C*_0_ × 100.

MDA content was quantified using kits (Suzhou Michy Biomedical Technology Co., Ltd., Suzhou, China) following the manufacturer’s instructions.

### 2.6. Extraction and Assay of Enzyme Activities

All steps for crude enzyme extraction were conducted at 4 °C. Five grams of tubers was homogenized with 20 mL of an extraction buffer. This mixture was then centrifuged at 10,000× *g* for 20 min, and the supernatant was reserved for enzyme assays.

The activities of α-amylase, SOD, and CAT were determined utilizing kits supplied by Suzhou Michy Biomedical Technology Co., Ltd., Suzhou, China, in accordance with the manufacturer’s instructions.

### 2.7. Statistical Analysis

Data were statistically examined using one-way ANOVA in SPSS 22.0 (SPSS Inc., Chicago, IL, USA). Each experiment followed a completely randomized design and was replicated three times. Duncan’s test (*p* < 0.05) determined significant variances. All results are presented as mean ± standard deviation (SD). Graphical representations were generated using OriginPro 2019b (Origin Lab., Micro Cal, Northampton, MA, USA).

## 3. Results and Discussion

### 3.1. Weight Loss Rate and Tuber Decay Incidences

Figure 1A presents a detailed analysis of the impacts of diverse treatments on the weight loss rates of potatoes during their late dormancy phase. As the storage duration increased, a consistent trend of tubers undergoing weight loss was observed. Notably, after 30 days of storage, the GA3 treatment manifested a weight loss rate that was 15.14% higher compared to the water treatment. Conversely, treatments employing carvone, ABA, and variable-temperature modulations exhibited weight loss reductions of 12.25%, 12.13%, and 11.63%, respectively, when contrasted with the control. These data articulate the pronounced efficacy of carvone, ABA, and variable-temperature treatments in curtailing weight loss throughout storage, with the carvone treatment emerging as the most impactful.

The prolonged storage of potatoes invariably results in weight loss, a phenomenon attributed to numerous physiological and metabolic changes that occur during their dormant state. During the dormancy phase, the metabolic rate of the seed decreases, resulting in energy conservation. However, the moment this dormancy is interrupted, the seeds spring back to life, leading to moisture evaporation and subsequent weight loss [25]. The GA3 treatment appeared to stimulate this reactivation, thereby causing a more pronounced weight loss. In contrast, the treatments involving carvone, ABA, and variable temperature seemed to prolong the dormancy phase, leading to attenuated weight loss [26]. It has been reported that carvone treatment is pivotal in sustaining dormancy, regulating bud length, and thwarting membrane lipid peroxidation in tubers. It mitigates weight loss and decay rate in seed potatoes during storage, preserves elevated SOD and CAT activity, diminishes MDA accumulation and the scope of membrane lipid peroxidation, delays physiological aging, augments ABA levels, and extends the dormant period in seed potatoes [27]. Variable-temperature treatment, by reducing the respiration rate and water evaporation, helps in minimizing the weight loss of fruits and vegetables [28].

Figure 1B illustrates a comprehensive analysis of the differential effects of treatments on the decay rate of aeroponic tubers as they traverse the late dormancy period. As storage duration increased, there was a consistent increase in the tuber decay rate. An interesting observation was that tubers subjected to the ABA treatment exhibited 0 decay at the 5-day mark but exhibited a peak in decay rate by day 15. In a similar vein, both the carvone and GA3 treatments reached their maximum decay rate at the 25-day mark (3.33%) before stabilizing. The overall decay rate reached a maximum of 4% by the 25th day and maintained this level thereafter. Importantly, all treatment modalities, which include carvone, ABA, GA3, and variable temperature, registered decay rates that were lower than the control, highlighting their protective effectiveness in tuber preservation. Of these, the ABA treatment emerged as particularly impressive, demonstrating superior decay prevention properties.

Increasing decay rates during storage are likely attributable to microorganisms thriving under the specific storage conditions. Even as storage extends, decay persists, potentially arising from increased metabolic activities in the seeds once dormancy is broken [29]. The treatments with carvone and GA3 appeared to reduce the decay rates, potentially by inhibiting microbial proliferation and simultaneously preserving the inherent physiological state of the tubers. Carvone, a compound extracted from the essential oil of coriander seeds, is recognized as an environmentally sustainable antibacterial agent. Notably, carvone displays formidable antibacterial activity against pathogens commonly encountered during potato storage, notably *Fusarium sulphureum*, effectively staving off afflictions like dry rot in stored potatoes [30,31].

### 3.2. Sprout Length and Germination Rate

Figure 2 presents a comprehensive analysis of how the different treatments impacted the sprout length and germination rates of aeroponic primary seeds during their late dormancy phase. As the storage period extended, a consistent increase in sprout length of these primary seeds was observed. Notably, the GA3 treatment resulted in the longest sprouts, reaching a maximum length of 14.24 mm after a span of 30 days. In contrast, the carvone treatment resulted in the shortest sprouts, measuring a mere 7.53 mm.

Following their storage phase, potato seeds transition into a significant dormancy period, characterized by a reduced metabolic rate, which is not conducive for bud proliferation. It is plausible that GA3 treatment acts as a catalyst in bud growth, spurring both cell division and elongation. Conversely, carvone treatment, with its distinct properties, might act as a deterrent to bud growth. When carvone acts on seed potatoes, it stimulates ABA synthesis by inhibiting HMG-CoA reductase and hinders IAA and GA3 synthesis, thereby inhibiting seed potato germination. The removal of carvone increases HMG-CoA reductase activity, inhibits ABA content, and increases IAA and GA3 content, resulting in the termination of seed potato dormancy [32]. Carvone treatment significantly inhibits MDA accumulation in seed potatoes. Carvone treatment significantly increases SOD and CAT activity in seed potatoes and effectively prevents the accumulation of reactive oxygen species [14]. Therefore, carvone treatment helps maintain dormancy, controls bud length, and prevents membrane lipid peroxidation in seed potatoes.

Within the initial 15 days of observation, the germination rates across all treatment paradigms showed an increase, subsequently stabilizing between the 15th and 30th day. The ABA treatment was the most effective, achieving a germination rate of 97.33%. The other treatments, though not matching the efficacy of ABA, still demonstrated notable results, with carvone and GA3 treatments recording 94%, the control hovering at 93.33%, and the variable-temperature treatment registering 94.67%. In this ensemble of treatments, the efficacy of the ABA treatment in germination efficiency was unparalleled.

The increasing germination rates can potentially be ascribed to the seeds gradually overcoming their dormancy state during their storage, leading to the weakening of their intrinsic dormancy mechanisms. ABA has been well documented in the botanical literature as an agent that inhibits seed germination. Positioned as a chief instigator of dormancy, multiple studies have expounded on ABA’s key role in perpetuating dormancy and the subsequent decline in its levels as dormancy is released [33]. Further research into the identification of ABA inhibitors and mutants in ABA synthesis has solidified its reputation as a dormancy-enhancing compound. However, it is significant that this dormancy-inducing effect of ABA can be neutralized by GA3, offering a feasible approach to inhibit germination and thereby mitigate the risk of tuber afflictions during their storage phase [34]. The intrinsic dynamics of ABA play a key role in the dormancy process of tubers. As the tuber’s formation process unfolds, there is a surge in its endogenous ABA concentrations, which, in turn, inhibits tip growth. However, once the dormancy phase is terminated and the stem tip begins its growth, there is a sharp decrease in ABA levels. Consequently, ABA’s significant influence is most palpable in the ex vivo dormancy of potato tubers. As this dormancy phase reaches its conclusion, there is a marked decrease in tuber ABA concentrations, leading to a shift in the ABA-GA ratio, which acts as the primary catalyst for tuber germination. Increasing endogenous ABA levels through external application emerges as an optimal strategy to curtail the germination rates of potatoes during their storage phase.

As the storage phase progresses, there is a gradual depletion in the seed’s internal reserves of ABA, which, in turn, reduces its germination-suppressing effect, leading to an increase in germination rates. GA3, with its distinctive attributes, might amplify these germination rates by stimulating cell division and growth, thereby counteracting the inhibitory effect of ABA [35,36]. Research into the effect of carvone on seed potato bud meristems have unveiled that in the absence of any external treatment, these seed potatoes, owing to their inherent apical dominance, prioritize the activation of their apical bud. However, a mere 2-day exposure to carvone treatment initiates a deterioration sequence in the apical meristem end and its vascular tissue, culminating in its necrosis within a 5–7-day window. This phenomenon might either spring from carvone’s modulating effects on hormonal dynamics or its hydrophobic molecular structure, which influences its biological activity. The latter could inflict damage on the cell membranes of meristematic tissue, leading to necrosis in the apical bud, while its axillary counterpart remains under the influence of hormonal regulation and preserves its sprouting vitality. A hiatus of three days post-treatment witnesses a waning of carvone’s influence, likely attributable to shifts in the endogenous hormonal spectrum within the seed potatoes via the methyl hydroxybutyric acid pathway, which, in turn, promotes the growth of the axillary buds [37]. Temperature shifts can influence the production and degradation of plant hormones, notably gibberellin and abscisic acid. These hormones are pivotal in governing seed dormancy and germination. Optimal temperatures can enhance the synthesis of GA3. Some seeds, after undergoing cold stratification (a process simulating winter conditions), produce higher amounts of GA3, facilitating their germination [38]. On the other hand, elevated or non-ideal temperatures might amplify the production of abscisic acid, reinforcing seed dormancy [39]. In contrast, appropriate temperature treatments, like cold stratification, can promote the breakdown or diminish the synthesis of abscisic acid, paving the way for seed germination.

### 3.3. Electrical Conductivity and Malondialdehyde Content

Figure 3A presents a comprehensive analysis of the varying treatments’ implications on the relative electrical conductivity of aeroponic primary seeds during their late dormancy phase. Throughout the storage phase, seeds subjected to various treatments demonstrated a notable trend: a temporary decrease in electrical conductivity, followed by a gradual increase. Notably, between the 10th and 25th days, the carvone treatment consistently resulted in the lowest conductivity levels, demonstrating better results than other treatments. Yet, by the end of the 30-day observation period, all treatments exhibited electrical conductivities that exceeded those of the control, suggesting some level of cell membrane deterioration in the primary seeds. Within this group, the carvone treatment appeared to cause the least cellular damage.

Electrical conductivity is a critical metric, especially for determining the nutrient balance and subtle variations in osmotic pressure in plants cultivated hydroponically. It serves as an essential indicator for measuring the integrity and permeability of cell membranes. A surge in conductivity values typically indicates heightened permeability, suggesting increased cellular damage. In the context of significant dormancy, the metabolism of seeds undergoes a significant reduction, translating into attenuated electrical conductivity. The GA3 treatment might act as a catalyst, enhancing bud growth and thereby increasing electrical conductivity [35]. Conversely, carvone might play the role of an inhibitor, inhibiting bud growth and consequently leading to reduced conductivity.

Figure 3B systematically documents the variations in MDA content across the array of treatments during their storage period. MDA, a byproduct of lipid peroxidation, is a recognized indicator of membrane lipid peroxidation and associated cellular damage. Throughout the storage duration, the MDA content showed significant changes across treatments. By the critical 30-day mark, the carvone and ABA treatments showed increases of 6.73% and 0.96% from their initial values, respectively. In contrast, the CK, GA3, and variable-temperature treatments displayed decreases of 7.69%, 23.08%, and 7.69%, respectively. These results highlight the effectiveness of the GA3 treatment in inhibiting the increase in MDA content in primary seeds during storage, thereby reducing cellular damage.

The increase in MDA, a lipid peroxidation byproduct, indicates membrane lipid peroxidation, which is closely associated with cellular damage. An increase in MDA levels is a clear indicator of damage to cell membranes. The GA3 treatment appeared effective in reducing MDA content, potentially preserving the integrity of cell membranes and preventing the lipid peroxidation reactions that can significantly damage them [40]. Conversely, the carvone and ABA treatments seemed to increase MDA content, possibly due to their lower effectiveness in shielding cell membranes from the onslaught of lipid peroxidation. Previous research has illuminated that heat treatments can effectively reduce the accumulation of MDA in tubers while maintaining a relatively low electrical conductivity, thereby ensuring the stability and robustness of cell membranes. Temperature variations influence the fluidity of cell membranes, which can subsequently impact the activity of enzymes and the transport dynamics on the membrane. Seed germination is a process underpinned by the actions of numerous enzymes, including amylase and protease, which might remain dormant during seed dormancy. Variable-temperature treatments can activate these enzymes, facilitating the degradation of stored materials within the seed and supplying energy for germination. Certain seeds contain dormancy-associated proteins that could hinder germination. Fluctuating-temperature treatments can aid in breaking down these proteins, successfully terminating the dormancy [41].

### 3.4. α-Amylase, CAT, and SOD Activity

α-Amylase, playing a crucial role in starch metabolism, is a significant enzyme in seed physiology. Figure 4A presents a comprehensive analysis of temperature variations’ and chemical treatments’ influence on α-amylase activity in original seeds during their late dormancy phase. A noticeable pattern was observed across all treated original seeds throughout the storage span: an initial spike in α-amylase activity, which then tapers off. For instance, under the carvone treatment, α-amylase activity increased to a maximum of 5.78 mg/min/g by the 15th day, only to decrease by 4.29% from its initial level by the 30th day. The pattern of the variable-temperature treatment followed this trend, with a maximum of 8.51 mg/min/g by the 10th day and a subsequent 8.89% decrement from its outset by day 30. The treatments with water, ABA, and GA3 exhibited peak α-amylase activities on the 20th, 20th, and 5th days, showing increases of 59.51%, 40.49%, and 46.01% from their baselines by the 30th day, respectively. From a broader perspective, the trajectory of α-amylase activity declined as the storage window expanded, underscoring the imperative of maintaining a consistent amylase activity level for the entirety of seed storage.

α-Amylase is closely associated with starch metabolism. Its activity often increases during the initial stages of deep dormancy, likely owing to an accumulation of starch reserves within the seeds [42]. As the dormancy phase diminishes, the seeds utilize the stored starch for energy. However, this reliance diminishes over time, leading to a decrease in α-amylase activity. Carvone and GA3 treatments appear to modulate α-amylase activity in distinct ways: carvone potentially acts as an inhibitor, while GA3 might serve as a stimulant.

CAT, with its formidable role in disintegrating hydrogen peroxide into benign constituents like water and oxygen, is essential in the cellular antioxidant defense mechanism. Figure 4B examines the effects of temperature variations and chemical treatments on CAT enzyme activity in aeroponic original seeds during their late dormancy phase. A clear trend was observed across all treated original seeds throughout the storage tenure: CAT enzyme activities initially increased and then decreased. Strikingly, both the carvone and GA3 treatments achieved the highest levels of CAT enzyme activities, registering values of 112.73 and 93.69 μmol/min/g, respectively, by the 30th day. This translated into a marginal 0.95% increment for the carvone treatment and a notable 223.97% increase for the GA3 treatment from their starting points. In stark contrast, the control treatment reached its peak at 111.66 μmol/min/g by the 15th day, marking a robust 76.42% surge from its baseline by the 30th day. The water and ABA treatments reached their zeniths on the 5th and 20th days, with increments of 26.58% and 152.53% from their baselines by the 30th day, respectively. In an overall analysis, CAT enzyme activities under the variable-temperature, ABA, and GA3 treatments outpaced those under the water treatment, while the carvone treatment lagged slightly behind the control treatment.

CAT is crucial for protecting cellular structures against oxidative stress. An increase in CAT might be a strategic response to counterbalance elevated levels of hydrogen peroxide within cells after emerging from dormancy. Both carvone and GA3 treatments seem to stimulate CAT activity, potentially as a defense against cellular oxidative stress. Concurrently, the ABA and variable-temperature treatments might either minimally modulate or slightly reduce CAT activity.

Figure 4C details the effects of temperature variations and chemical treatments on SOD enzyme activity in aeroponic original seeds during their late dormancy phase. A notable finding is that SOD activity for all treatments peaked by the 15th day, with CK, carvone, ABA, and variable-temperature treatments registering values of 478.92, 462.5, 466.53, and 453.28 U/g, respectively. By the 30th day, carvone, water, ABA, and variable-temperature treatments showed reductions in SOD activity of 18.25%, 33.14%, 16.92%, and 32.91% from their baseline values, respectively. For the GA3 treatment, SOD activity initially decreased until the 20th day, after which a minor increase was observed, resulting in an 8.36% decrease from its baseline by the 30th day. At the end of the 30-day observation period, all treatments exhibited higher SOD activities compared to the CK.

SOD is fundamental in the plant’s antioxidant system, responsible for regulating the concentrations of superoxide anion radicals and hydrogen peroxide within cells, ensuring the cellular membranes are protected from damage. SOD, with its crucial role in the plant’s antioxidant system, reduces the harmful effects of superoxide anion radicals [43]. An increase in SOD activity might be a compensatory response against escalating superoxide anion radicals following the end of the dormancy phase. The carvone, ABA, variable-temperature, and GA3 treatments all appeared to bolster SOD activity, highlighting their potential effectiveness in combatting oxidative stress.

## 4. Conclusions

The complex dynamics of postharvest dormancy and germination in potato cultivation are crucial, having significant implications for both crop uniformity and overall yield. The effective management of germination during storage is essential. Not only does it mitigate potential losses, but it also ensures the sustainability and vitality of subsequent potato harvests.

This comprehensive study examined aeroponic original potato seeds, all of which were sourced from a single production batch of a family-operated venture. By using a combination of temperature control and various chemical treatments, our objective was to improve and standardize germination uniformity, priming the seeds for subsequent cultivation. The collected data provide strong evidence regarding the efficacy of the temperature variation regimen in modulating the weight attrition rate inherent to the primary potato specimens. Additionally, the ABA treatment was effective in reducing rot occurrences. Delving deeper, the treatments anchored around carvone, ABA, and variable temperature resulted in truncated germination lengths. Notably, the ABA treatment showed excellent performance, achieving a germination rate of 97.33%, closely followed by the variable-temperature treatment. Our findings suggest that while all treatments caused some cellular membrane damage, the carvone treatment was the least harmful, causing minimal damage. Conversely, the GA3 treatment effectively stymied the escalation of MDA content during the storage phase, acting effectively in cell membrane preservation. Furthermore, across the board, the enzymatic activities of α-amylase, CAT, and SOD followed a consistent trajectory: an initial surge, followed by a gradual tapering, maintaining a balanced enzymatic environment.

In summary, this research provides a deeper understanding of the mechanisms at play and offers practical insights for developing effective germination management strategies. Furthermore, it offers a blueprint for preserving the intrinsic quality of aeroponic potato original seeds during their extended dormant phase. The broader ramifications of these insights echo in the corridors of agricultural best practices, potentially leading to improvements in potato storage protocols and, ultimately, enhancing the yield and productivity of potato cultivation.

## Figures and Tables

**Figure 1 plants-12-03952-f001:**
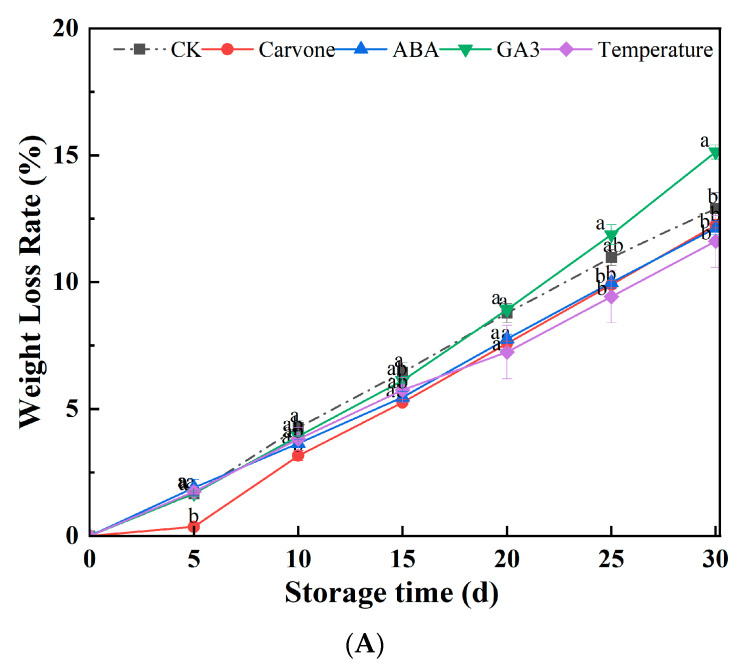
Effects of different treatments on weight loss rate (**A**) and tuber decay incidence (**B**) in mini-tuber potato seeds produced under aeroponics. Error bars represent the standard errors. Distinct letters indicate significant differences among each storage stage at *p* < 0.05. CK = control, Carvone = carvone, ABA = abscisic acid, GA3 = gibberellin, and Temperature = variable temperature.

**Figure 2 plants-12-03952-f002:**
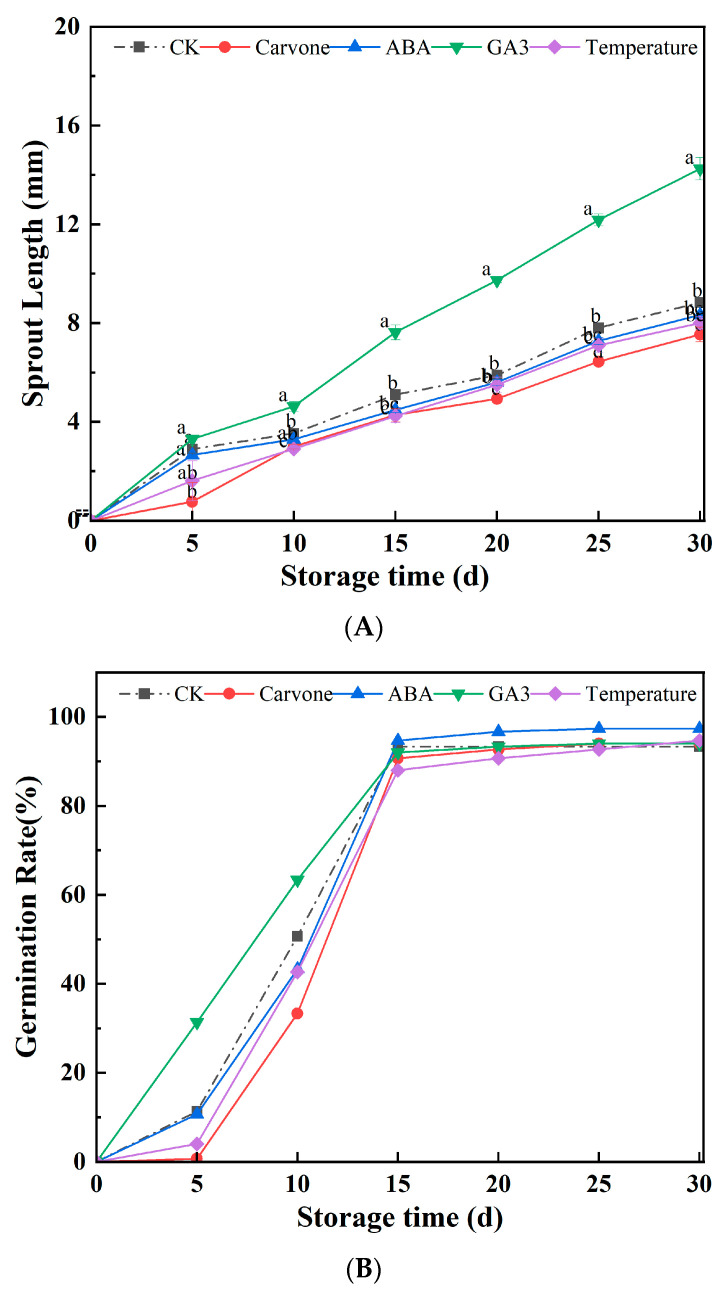
Effects of different treatments on sprout length (**A**) and germination rate (**B**) in mini-tuber potato seeds produced under aeroponics. Error bars represent standard errors. Distinct letters indicate significant differences among each postharvest stage at *p* < 0.05. CK = control, Carvone = carvone, ABA = abscisic acid, GA3 = gibberellin, and Temperature = variable temperature.

**Figure 3 plants-12-03952-f003:**
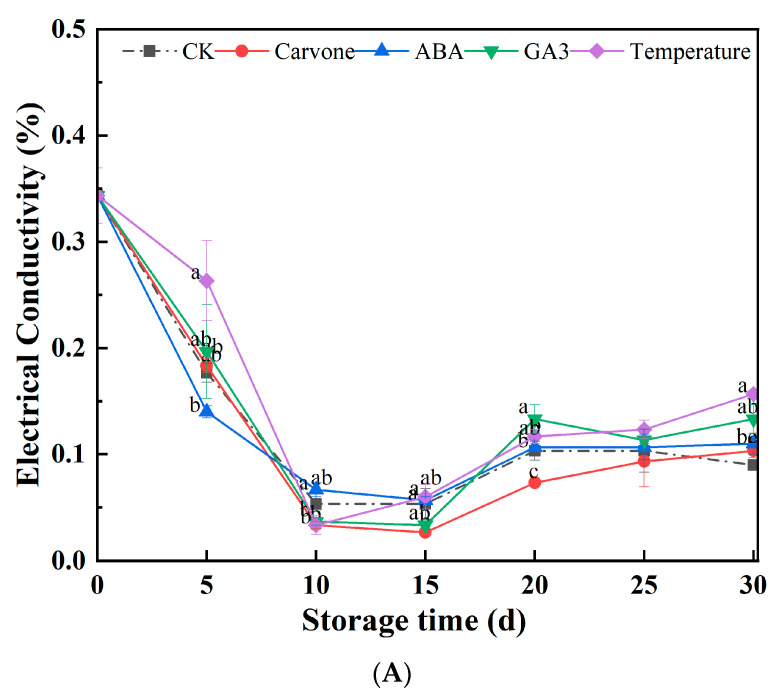
Effects of different treatments on electrical conductivity (**A**) and malondialdehyde content (**B**) in mini-tuber potato seeds produced under aeroponics. Error bars represent standard errors. Distinct letters indicate significant differences among each postharvest stage at *p* < 0.05. CK = control, Carvone = carvone, ABA = abscisic acid, GA3 = gibberellin, and Temperature = variable temperature.

**Figure 4 plants-12-03952-f004:**
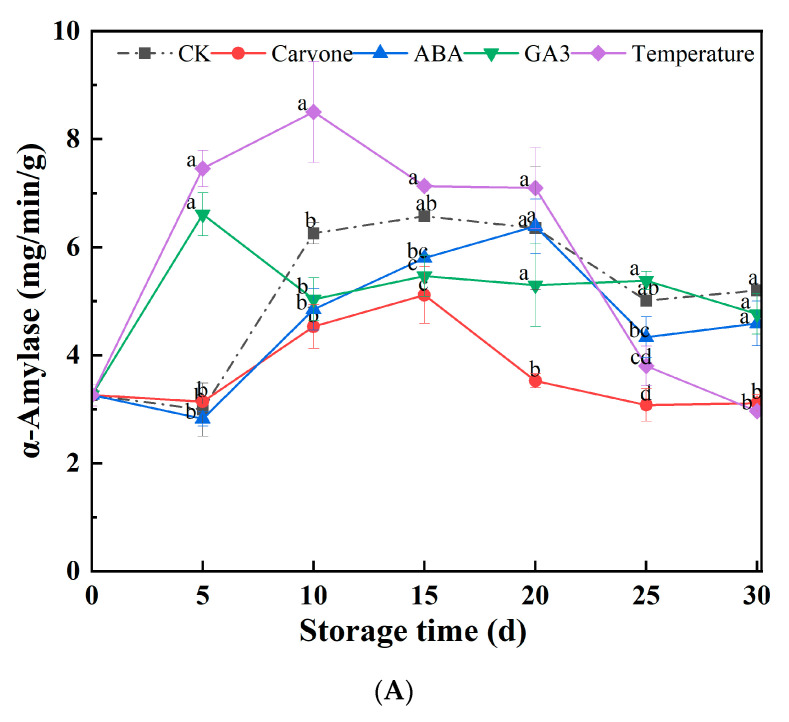
Effects of different treatments on activity of α-Amylase (**A**), CAT (**B**), and SOD (**C**) in mini-tuber potato seeds produced under aeroponics. Error bars represent the standard errors. Distinct letters indicate significant differences among each storage stage at *p* < 0.05. CK = control, Carvone = carvone, ABA = abscisic acid, GA3 = gibberellin, and Temperature = variable temperature.

## Data Availability

Data are contained within the article.

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
