# Peer review of "The Postharvest Application of Carvone, Abscisic Acid, Gibberellin, and Variable Temperature for Regulating the Dormancy Release and Sprouting Commencement of Mini-Tuber Potato Seeds Produced under Aeroponics"

_plants, 2023, doi:10.3390/plants12233952_

Round 1

Reviewer 1 Report

Comments and Suggestions for Authors

The manuscript presents a very complete study of the effect of growth regulators and temperature on dormancy, sprouting and enzymes involved in the initial stages of potato tuber seed development. The results contribute to the compression and management of tuber dormancy to produce this crop of great nutritional importance.

In general terms, the work has a robust design that provides significance to the results obtained, which integrate the results of conservation parameters of biological material and the biochemical processes of nutrient release and oxidative stress during the latency stage.

The inclusion of temperature as a treatment is a determining factor for the description of the process, however, it would be of great importance to show the combined effects of temperature and growth regulators, especially ABA, which would complement the comprehensive discussion of the effects. This suggestion is made in case the experimental design has contemplated a simultaneous interaction matrix of all treatments.

Another suggestion is that a general review be made of the writing of the manuscript, which presents an overly stylized style, which moves away from the technical approach of scientific communication and may limit the scientific community's interest.

Additional suggestions.

All graphs indicate a CK treatment, which I assume is the control. In this sense, it would be convenient to indicate in the text the description of this treatment explicitly.

Author Response

Dear Reviewer,

Thank you for your valuable comments and suggestions on our manuscript. We greatly appreciate your detailed feedback.

  1. You noted the importance of including temperature as a treatment factor and its combined effects with growth regulators, particularly ABA, for a comprehensive discussion of the effects. We acknowledge this insightful suggestion. In response, we have addressed your point regarding temperature being a variable treatment alongside chemical treatments. The manuscript has been revised accordingly to reflect this adjustment.
  2. You also recommended a general review of the manuscript's writing style, which was previously overly stylized and potentially limiting for scientific communication. We thank you for this observation and, in accordance with your advice, have thoroughly revised the language throughout the manuscript to ensure a more professional and technical tone. These changes have been clearly marked in the text for ease of identification.
  3. Regarding your additional suggestions about the CK treatment, we are grateful for your attention to this detail. As you correctly assumed, CK represents the control in our study. To clarify, we have now explicitly described the CK treatment within the manuscript.

Thank you once again for your constructive critique, which has undoubtedly strengthened the quality and clarity of our work.

Best regards,

Tiandi Zhu

Reviewer 2 Report

Comments and Suggestions for Authors

Dear Editor, the ms brings new information about the use of some plant growth regulators on dormancy control in mini-tuber potato seeds. All my comments are in the attached file. Special attention to the quality of the Figures must be considered, because the resolution of them are poor.

Author Response

Dear Reviewer,

Thank you sincerely for your insightful comments and recommendations regarding our manuscript. Your detailed review has been invaluable.

  1. We have now included the specific species name of the potato in the manuscript.
  2. The term "proved" has been updated to "demonstrate" to better convey the results.
  3. Detailed information about the reagents used has been added.
  4. All figures have been updated with complete labeling information.

Best regards,

Tiandi Zhu

Reviewer 3 Report

Comments and Suggestions for Authors

In general well written with a good standard of English even if there are sections that are written in slightly more poetic English than the normal fry unemotional stlye.

Section 2.5 ahs a slightly different style.  What were the minor alterations to the method of Xu and I would suggest deleting meticulously and the word accurately before gauged.  They are not needed and could imply that other work was rough and ready!

I do not find the graphs clear.  What is CK?  What does the Temperature mean?  I assume one of these is control but it is not clear.  Probably just another few words of explanation needed,, but this important  This is the main technical comment.

Bottom of page 6.  Segue is not a common word and I suggest you change it.

There are long paragraphs on page 7.  Remove "marches on" which is slang and put develops.  This paragraph has an unusual style with clout and a mere two day tryst and suggests that Shakespeare may be an influence!  Please use less poetical language. 

Comments on the Quality of English Language

The English is a little more poetical than normal but is still clearly understandable

Author Response

Dear Reviewer,

Thank you for your thorough review and constructive feedback on our manuscript. We have addressed your comments as follows:

  1. The minor alterations to the method of Xu in Section 2.5 have been clarified, and we have removed the unnecessary words "meticulously" and "accurately" before "gauged."
  2. To improve clarity, we have now provided explanations in the figures to indicate that "CK" represents the control and "Temperature" denotes the variable temperature treatment.
  3. The term "Segue" has been removed as your suggestion.
  4. The phrase "marches on" has been replaced with "develops" for a more precise and less poetical expression. Additionally, the stylistic elements involving "clout" and "a mere two day tryst" have been revised for a more straightforward presentation.
  5. We have reviewed and revised the overall language to ensure it aligns with a more technical and less poetical style.

We appreciate your valuable input, which has contributed to enhancing the quality and clarity of our manuscript.

Best regards,

Tiandi Zhu
